# FLOW-BASED IMPUTATION OF SMALL DATA

## ABSTRACT

Many challenges in the physical sciences can be framed as small data problems, where theoretical progress is hindered by the sparsity, low-dimensionality, and/or limited sample size of available empirical data compared to a physical system's numerous dynamical degrees of freedom. Developing trustworthy imputation methods for these datasets holds immense scientific importance. Normalizing flows are a promising model choice for imputation due to their ability to explicitly estimate sample likelihoods. However, research has shown that normalizing flows are often unreliable for out-of-distribution (OOD) detection in high-dimensional settings, which undermines their trustworthiness for imputation tasks. In contrast, low-dimensional settings provide opportunities to tractably evaluate and mitigate likelihood estimation errors, revealing strategies to reduce or eliminate specific error modes. We focus on the most stringent assumption in normalizing flows: diffeomorphism between the target and base distributions. This assumption introduces two distinct error modes, which we identify and address through a simple and effective strategy. Our approach significantly enhances the trustworthiness of normalizing flows for imputation in small data problems.

Many frontiers of physical science can be characterized as small data problems in which theoretical progress is hindered due to the high costs and/or infeasibility of conducting experiments or making observations (Hossenfelder, 2018; Coveney et al., 2016). We define *small data* as datasets that are low-dimensional, sparse, and/or limited in sample size relative to the complexity of the underlying probability distributions from which they are sampled. Examples of scientific small data problems include the dependency of jets from defects in shocked metal surfaces on defect geometry, shock dynamical parameters, and material properties, and other variables (Kaiser et al., 2024) as well as the dependency of measures of oceanic turbulence on local bathymetry, distance from the equator, stratification, depth, surface wind stresses, tidal forcing, and other variables (Gregg et al., 2018; Moum, 2021). Trustworthy yet flexible imputation could not only improve predictive capabilities and uncertainty quantification for problems like these but also inform the design of future experiments and observations.

Normalizing flows are an appealing tool for imputation because they offer exact likelihood estimation and efficient sampling (Papamakarios et al., 2021; Kobyzev et al., 2020; Prince, 2023). We refer a *likelihood* as the probability or density of a sample under a distribution, as is common in the literature (Zhang et al., 2021), even though the formal definition of the term refers to a measure of how probable a set of parameters is, given some observed data. Flow trustworthiness, and therefore trustworthy flow-based imputation, hinges on whether the *model target distribution* (i.e., the model representation of the true target distribution) accurately represents the true target distribution. The *true target distribution* is the probability distribution underlying the training data samples which is usually unknown.

A major impediment to flow trustworthiness is the issue that, when flows and other deep generative models are applied to high dimensional datasets, they frequently assign higher likelihood to out-of-distribution (OOD) samples than to in-distribution samples (Caterini & Loaiza-Ganem, 2022; Zhang et al., 2021; Nalisnick et al., 2019). This phenomena obfuscates OOD sample detection (Yang et al., 2024; Fang et al., 2022; Zhang et al., 2021). OOD sample detection, hereafter *OOD detection*, is the task of identifying "whether the test example is from a different distribution from the training data" (Hendrycks & Gimpel, 2017). OOD detection methods distinguish OOD samples as low likelihood samples under the model target distribution (Bishop, 1994).

One explanation for model assignment of high likelihoods to OOD samples is the *typical set hypothesis*, which states that the relevant definition of OOD should instead be determined by the typical set of a distribution. A distribution's *typical set* is the set that contains the majority of the probability mass but not necessarily the highest densities or probability points (Choi et al., 2018; Nalisnick et al., 2020; Wang et al., 2020; Morningstar et al., 2021). Other plausible explanations for the frequency of this phenomena include model estimation errors (Zhang et al., 2021) and impossibility proofs that show that OOD detection is not possible under certain conditions, such as in- and out-distribution overlap (Fang et al., 2022).

OOD detection is a more tractably assessed problem in low dimensions than in high dimensions. Here, we directly assess the impact of errors that are produced by an axiomatic constraint of normalizing flows, the diffeomorphism of the model target and base distributions, in two dimensions. While other aspects of normalizing flows engender errors (Kirichenko et al., 2020; Zhang et al., 2021), we find that the diffeomorphism constraint can induce significant OOD flow likelihood and we provide a simple mitigation strategy to minimize OOD flow likelihood. We expect that the assessment of errors produced by the diffeomorphism constraint presented here is contribute to model target distribution error for higher dimensional data because the constraint applies to all flow models.

## 1 OVERVIEW OF NORMALIZING FLOWS

Normalizing flows are a class of deep generative models that attempt to model the true target distribution by mapping target samples to a prescribed *base distribution*. Flows approximately satisfy the change of variables equation that relates two densities,

$$\int_{\mathbb{S}_x} p_x(\mathbf{x})d\mathbf{x} = \int_{\mathbb{S}_x} p_z(f^{-1}(\mathbf{x}))\Big| \det \frac{\partial f^{-1}(\mathbf{x})}{\partial \mathbf{x}}\Big| d\mathbf{x}, \tag{1}$$

where the real samples $\mathbf{x} \in \mathbb{R}^d$ have dimension $d$ under support $\mathbb{S}_x$ are bijectively mapped to real samples $\mathbf{z} \in \mathbb{R}^d$ by the mapping function $f$,

$$\mathbf{x} = f(\mathbf{z}). \tag{2}$$

Usually, a single isotropic Gaussian distribution is chosen for the base distribution $p_z$, hence the flow *normalizes* the target density $p_x$ as it is transformed through successive layers.

Normalizing flow architectures consist of many compositions of Equation 1. They relate the unknown target density is $p_x$ to the prescribed base density is $p_z$ through

$$\int_{\mathbb{S}_x} p_x(\mathbf{x})d\mathbf{x} = \int_{\mathbb{S}_x} p_z(f_1^{-1}(...f_n^{-1}(\mathbf{x})...))\Big| \prod_{i=1}^{n} \det \frac{\partial f_i^{-1}(\mathbf{x})}{\partial \mathbf{x}}\Big| d\mathbf{x}, \tag{3}$$

where $n$ is the number of layers and the composed mapping functions are

$$\mathbf{x} = f_1(...f_n(\mathbf{z})...). \tag{4}$$

This is the origin of the name *flows*, which refers to the sequence of composed mappings, or *layers*, which expand or contract one density to form the other (Rezende & Mohamed, 2015). Mapping functions are usually simple in form, such as the scale and shift functions of RealNVP (Dinh et al., 2017), and deep neural networks learn the mapping functions.

The training objective is to use the target samples $\mathbf{x}$ to learn $f$ while satisfying Equation 3. Flows are often trained by maximum likelihood estimation of the target samples under the model (Atanov et al., 2019) or, equivalently, by evaluating the Kullback-Leibler divergence between the target density and the base density (Papamakarios et al., 2021).

Naive computation of the Jacobian determinant scales with $\mathcal{O}(n_s!)$, where $n_s$ is the number of samples in $\mathbf{x}$. Therefore, Dinh et al. (2014) proposed a 'masking' scheme for the mapping function for half of the $\mathbf{x}$ samples such that the Jacobian is lower or upper diagonal and its determinant can be tractably computed by merely multiplying the diagonal elements. The lost information of the masked mapping functions are dealt with by adding another layer with the opposite half of the mapping functions masked. By construction, the Jacobians of the two, oppositely masked, layers are equivalent to the LU decomposition of the Jacobian of a single unmasked layer. Hence, they

are referred to as *coupling layers* because pairs of oppositely masked layers are required to prevent information loss.

The mapping functions $f_i$ must be bijective, continously differentiable, and their inverses must be continously differentiable to satisfy Equation 3. Together, these properties impose the constraint that the model target and base distributions are diffeomorphisms (Kobyzev et al., 2020). A *diffeomorphism* is a homeomorphism that is also differentiable and has a differentiable inverse, and a *homeomorphism* is a continuous, bijective function between two topological spaces that has a continuous inverse (Munkres, 2000).

## 2 ERRORS PRODUCED BY THE DIFFEOMORPHISM CONSTRAINT

We identify two forms of model target distribution error that are produced by the diffeomorphism constraint. Both errors are

1. sources of spurious patterns of OOD likelihood and,

2. produced by topological inequivalences between the approximately or exactly totally disconnected regions of the supports of model target distribution and base distribution.

Figure 1 shows both types of error. Figure 1a shows a true target distribution characterized by a set three support modes,

$$\mathbb{S}_x = \mathbb{S}_{t*} = \text{Supp}(p_x) = \{\mathbb{S}_1, ..., \mathbb{S}_{m_{t*}}\}, \tag{5}$$

where the $t$ subscript denotes 'target distribution' and the $*$ denotes either the true target distribution. Each mode of the support is defined

$$\mathbb{S}_i = \{\mathbf{x}_i \in \mathbb{R}^d : p_x(\mathbf{x}_i) > 0\}, \tag{6}$$

$$\mathbf{x} = \{\mathbf{x}_1, ..., \mathbf{x}_{m_{t*}}\}, \tag{7}$$

$m_{t*}$ is the number of modes in the true target distribution, which is generally unknown for most scientific datasets. $m_b$ is the number of modes in the base distribution and $m_t$ is the number of modes in the (trained) model target distribution, and

We define a *mode* as an approximately or exactly totally disconnected region (i.e., manifold) of the support of a distribution. *Totally disconnected* means that the regions do not topologically overlap (Munkres, 2000) and *approximately* means the non-zero probability regions are separated by regions of approximately zero density or probability. Regions of zero probability are outside of the support of the probability density function by definition (Billingsley, 1995). Therefore, the dark regions of the true target distribution in Figure 1a (denoted by the modes defined as subsets of the support $\mathbb{S}_{t*} = \{\mathbb{S}_1, \mathbb{S}_2, \mathbb{S}_3\}$) are in-distribution and the surrounding light regions are out-distribution (either exactly or approximately zero probability or density).

### 2.1 THE MODAL DIFFEOMORPHISM CONJECTURE

The errors discussed in this paper are all derived from the empirically-informed conjecture that, if the model requires that the map $f$ is diffeomorphic, where

$$f : \mathbf{x} \to \mathbf{z}, \tag{8}$$

then the model also requires that the modal map $f_{\mathbb{S}}$ be diffeomorphic, where

$$f_{\mathbb{S}} : \mathbb{S}_x \to \mathbb{S}_z, \tag{9}$$

for the finitely bounded supports $\mathbb{S}_x$ and $\mathbb{S}_z$ with finitely bounded modes. Thus, the number of model base modes is equal to the number of model target modes,

$$m_b = m_t, . \tag{10}$$

We refer to Equation 10 as the *modal diffeomorphism* conjecture.

| True target distribution | Prescribed base distribution | Forward mapped target samples | Model target distribution |
|---|---|---|---|

Figure 1: The two types of error are produced by modal diffeomorphism ($m_b = m_t$). $a$ shows a 2D multi-modal true target distribution, while $b, c, d$ show a *target mode error*, which occurs when the prescribed base distribution has fewer modes than true target distribution. The model adds spurious filaments of OOD likelihood to satisfy modal diffeomorphism (Equation 10). $e, f, g$ show *base mode error*, which occurs when the model fails to map the target samples $\mathbf{x}$ to all of the prescribed base distribution modes. Sampling the base distribution then generates spurious and randomly placed blobs of OOD likelihood in the model target distribution (Figure $g$) corresponding to the omitted base distribution mode(s) (i.e., the red circle in $f$).

## 2.2 TARGET MODE ERROR

*Target mode error* occurs when the model target distribution has fewer modes than the true target distribution,

$$m_t < m_{t*}. \tag{11}$$

It occurs when the true target distribution is multi-modal ($m_{t*} \geq 2$) and $m_{t*}$ is greater than the number of base distribution modes $m_b$,

$$m_b < m_{t*}, \tag{12}$$
$$m_b \geq 1. \tag{13}$$

The true map of the true target distribution modes onto the prescribed base distribution modes,

$$f_{\mathbb{S}*} : \mathbb{S}_{t*} \to \mathbb{S}_b, \tag{14}$$

is a *surjective map* because two or more modes of $\mathbb{S}_{t*}$ are mapped onto a smaller number of base modes in $\mathbb{S}_b$. Thus, the modes of the true target distribution and the base distribution are not bijective. And yet, the model map of model target distribution modes onto the model base distribution modes satisfies modal diffeomorphism (Equation 10),

$$f_{\mathbb{S}} : \mathbb{S}_t \to \mathbb{S}_b. \tag{15}$$

The flow inserts OOD likelihood filaments to connect model target distribution modes to satisfy Equation 10. This is shown in Figure 1c, where the model target distribution resembles the shape of the true target density but OOD filaments of likelihood have been spuriously added by the flow to enforce modal diffeomorphism.

## 2.3 BASE MODE ERROR

At first glance, it would appear that the target mode error described in the previous section could be eliminated by simply matching the number of prescribed base distribution modes to the number

Figure 2: The analytical multi-modal toy distribution. The dark regions show uniform non-zero probability density and the white regions are out of distribution. Each square is unit width and they are spaced such that the expectation is zero and the standard deviation is unity.

of true target distribution modes. The problems with this approach are a) the number of true target distribution modes is generally unknown *a priori* and b) that the flow can fail to map all of the target samples to all of the base distribution modes. Problem b) produces another type of error we refer to as base mode error. *Base mode error* occurs when the model maps the target samples to only a subset of the prescribed base modes,

$$m_{b\rightarrow} < m_b, \tag{16}$$

where the number of base modes that the model forward maps the target samples onto is denoted $m_{b\rightarrow}$. This is shown in Figure 1f, where the target samples have only been mapped to three out of four prescribed base modes. This creates spurious blobs of OOD likelihood in the model target distribution (Figure 1g) when the base distribution (Figure 1e) is randomly sampled.

## 3 EMPIRICAL RESULTS

### 3.1 A SIMPLE ARCHITECTURE

To empirically investigate target mode error and base mode error we use the RealNVP (Dinh et al., 2017) normalizing flow included in the Keras API (Chollet et al. (2015), https://keras.io/examples/generative/real_nvp/) with Tensorflow 2.4.1 and Tensorflow Probability 0.12.1. We trained all models for 300 epochs with a batch size of 256 samples and used a learning rate of $10^{-4}$. The following architectural details and hyperparameters were optimized for 2D toy distributions by trial and error. Each coupling layer contained scale and shift functions composed of five densely connected layers with 256 parameters each and ReLU activations, except for the output layer activations which were linear and hyperbolic tangent, respectively. Twelve coupling layers were used. The parameters were initialized with GlorotUniform initialization (Glorot & Bengio, 2010) and $l_2$ reglarized with a regularization coefficient of $10^{-2}$. The flows were trained using the batch mean of the negative log likelihood (NLL) to compute the loss,

$$\text{NLL}(\mathbf{x}) = -\log p_z(f_\theta^{-1}(\mathbf{x})) - \log\left|\det\frac{\partial f_\theta^{-1}(\mathbf{x})}{\partial \mathbf{x}}\right|, \tag{17}$$

$$\text{loss} = \frac{1}{n_b}\sum_{i=1}^{n_b}\text{NLL}(\mathbf{x}_i), \tag{18}$$

where $n_b$ is the batch size and $f_\theta$ represents the all of the composed layers in Equation 3, such that $f_\theta = f_1(...f_n(\mathbf{z})...)$.

We trained models with different numbers of base distribution modes ranging from one to eight ($1 \leq m_b \leq 8$). Each base distribution mode was an isotropic Gaussian distribution with a standard deviation of 0.05 and the mean of all Gaussian modes was a distance of at least 1 from the mean of all other modes for $m_b > 1$ cases. The ratio of the standard deviation to the distance between mode centers of 1/20 was sufficient for approximating each Gaussian as a separate mode (approximately totally disconnected) with regions of approximately zero probability separating the modes.

## 3.2 A MULTI-MODAL TOY TARGET DISTRIBUTION

To quantitatively evaluate target mode error and base mode error we need an analytical true target distribution with multi-modal support. A two dimensional true target distribution is ideal for visualization purposes. An analytical target distribution of four unit squares of uniform probability density ($m_{t*} = 4$) is sufficient for these ends, shown in Figure 2. The squares are spaced such that the mean of the distribution (including all modes) is zero and the standard deviation is unity in all dimensions.

## 3.3 TARGET MODE ERROR

Our empirical results suggest that target mode error is a systematic error (i.e., a deterministic error, Wheeler et al. (1996)): It occured every time the number of true target distribution modes exceeded the number of prescribed base distribution modes (Equation 12), regardless of weight initialization. We observed target mode errors in the form of connective filaments of OOD likelihood in the model target distribution for all $m_b < m_t = 4$ and we trained 36 models for each $m_b = \{1, 2, 3\}$.

Figures 3a, 3b, and 3c show the OOD likelihood filaments that connect the model target distribution to satisfy modal diffeomorphism (Equation 10) for $m_b = \{1, 2, 3\}$. There are no OOD likelihood filaments connecting the squares in Figure 3d because the number of prescribed base modes matches the number of modes in the true target distribution ($m_b = m_{t*}$) and the model successfully mapped all target samples to all base modes ($m_{b\rightarrow} = m_b$). Figures 3a, 3b, 3c, and 3d also show the residual $\rho(\mathbf{x})$, which we define as

$$\rho(\mathbf{x}) = |L_{x*}(\mathbf{x}) - L_x(\mathbf{x})|, \tag{19}$$

where $L_{x*}(\mathbf{x})$ is the true target distribution's likelihood evaluated at $\mathbf{x}$ and $L_x(\mathbf{x})$ is the model target distribution's likelihood evaluated at $\mathbf{x}$.

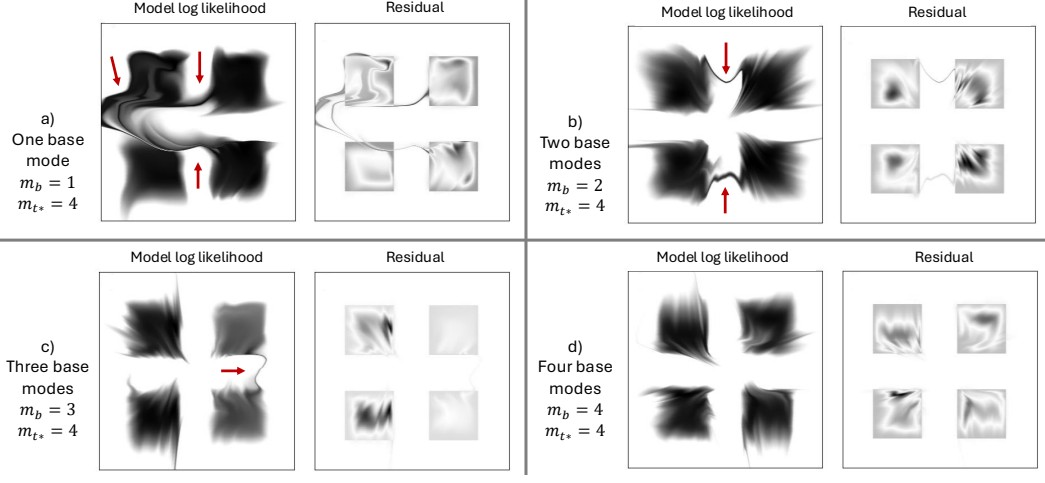

Figure 3: Target mode error examples for the model described in Section 3.1 applied to the (analytical) true target distribution shown in Figure 2. The plots on the left of $a, b, c, d$ show the model target distribution after training in the form of contours of the model log likelihood. Low model log likelihood (approaching $-\infty$) is shown with white regions (i.e., the model out-distribution) and high model log likelihood is shown with the gray and black regions (i.e., the model in-distribution). The plots on the right of $a, b, c, d$ show the residual for each case, as defined in Equation 19. The filaments of OOD likelihood added by the model to preserve modal diffeomorphism are indicated by the arrows.

## 3.4 BASE MODE ERROR

The empirical results suggest that base mode error a) is a random error (Wheeler et al., 1996) and b) can occur for $m_b < m_{t*}$, $m_b = m_{t*}$, or $m_b > m_{t*}$. The empirical results also suggest that base mode error becomes more likely as the ratio of prescribed base modes to true target modes $m_b/m_{t*}$ increases and that it occurs for all $m_b > m_{t*}$. Figure 4 shows the mapping of target samples $\mathbf{x}$

(first column) to the prescribed base modes (second column) for four different cases of base mode error. The omitted base modes are mapped into a single high likelihood randomly-placed mode of the model target distribution (third and fourth columns). The diffeomorphism constraint, as well as model accuracy, appears to break down when $m_b > m_{t*}$.

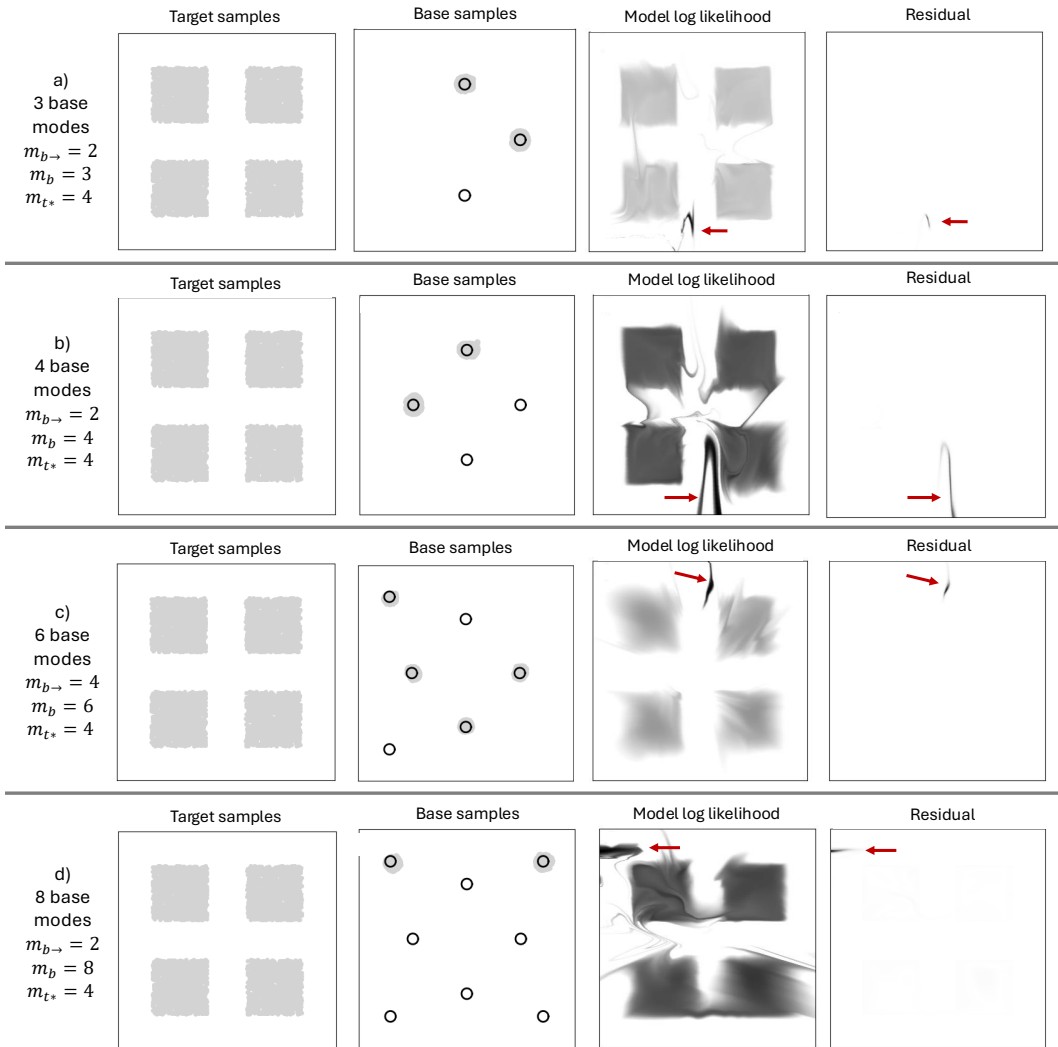

Figure 4: Four different cases of base mode error. $a$ is an example of base mode error for fewer prescribed base distribution modes than the number of true target distribution modes ($m_b < m_{t*}$). $b$ is an example of base mode error for an equal number of prescribed base distribution modes and true target distribution modes ($m_b = m_{t*}$). $c$ and $d$ are examples of base mode errors for a greater number of prescribed base distribution modes than true target distribution modes ($m_b > m_{t*}$). The target samples are shown as gray dots in the first column. The circles shown in the second column correspond to the 95% confidence interval (approximately two standard deviations) for each prescribed Gaussian distribution, and the empty circles are the base modes that have no mapping to the target samples. The spurious blobs of high OOD likelihood are indicated with arrows.

## 4 ERROR MITIGATION

A simple criterion for ensuring the optimal fit of the model target distribution to the true target distribution is to vary the number of base distribution modes and select the number of base modes of the model with the highest number of base distribution modes that also does not omit any base

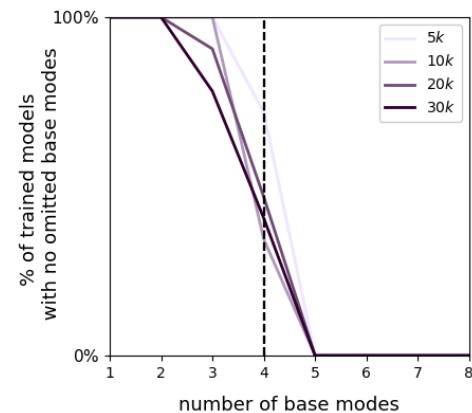

Figure 5: The percentage of trained models with no omitted base modes ($m_{b\rightarrow} = m_b$) as a function of the number of base modes. 36 models where trained for each base mode and for each dataset size ($n_s = \{5, 10, 20, 30\} \cdot 10^3$ samples).

modes, $\mathcal{M}(m_{b\rightarrow} = m_b)$,

$$m_b = \arg\max_{m_b} \mathcal{M}(m_{b\rightarrow} = m_b). \tag{20}$$

Models that fit this criterion do not suffer target mode error nor base mode error. We implemented the criteria by defining an omitted base mode as any prescribed base mode that had exactly zero target samples mapped to within its first standard deviation. Figure 5 shows the percentage of models with no omitted base modes (i.e., the models $\mathcal{M}(m_{b\rightarrow} = m_b)$) as a function of the number of base modes. It shows that the architecture used here has no base mode error when $m_b = m_{t*}$ for For $m_b \geq 5$ all models have omitted base modes, $\mathcal{M}(m_{b\rightarrow} < m_b)$.

Figure 6 shows the averaged loss over the final 50 epochs of training, the integrated OOD residual, and the OOD ratio for all trained models ('all results'), only the models that omitted no base modes ('no base mode errors'), and the models with base mode error ('omitted base modes'). The integrated OOD residual (second row of Figure 6) is defined as

$$\text{integrated OOD residual} = \int_{-\infty}^{\infty} \rho(\mathbf{x}) \backslash L_{x*}(\mathbf{x}) \, d\mathbf{x}, \tag{21}$$

where the backslash represents set substraction (i.e., the integrated OOD residual is zero within the black true target distribution regions of Figure 2). If the support of the model target distribution perfectly matches the support of the true target distribution (Figure 2), then the integrated OOD residual is exactly zero. The OOD ratio is the ratio of the number of OOD samples generated to the number of samples in the training dataset, $n_s$, for $n_s$ generated samples. If all generated samples are OOD (all generated samples lie within the white areas of Figure 2), then the OOD ratio is unity.

The results shown in Figure 6 have two important implications. First, the loss and the OOD ratio are not as good a measure of model target distribution error as the integrated OOD residual, even if we only examine the 'no omitted base mode' trends (middle column). The loss is batch averaged over $n_b$ samples and therefore in- and out-distribution likelihoods tend to be averaged together. The OOD ratio effectively represents only relatively high likelihood OOD regions because that is where most generated samples will occur, and therefore tends to insufficiently represent low likelihood OOD regions. The integrated OOD likelihood is a much better measure of model target distribution accuracy because it excludes the true target distribution and only quantifies OOD likelihood. Second, Figure 6e shows a clear downward trend as $m_b$ increases to match $m_{t*}$. The integrated OOD residual declines as the number of base modes is increased to match the number of true target distribution modes because fewer filaments of OOD are added to satisfy modal diffeomorphism. The smallest amounts of integrated OOD residual (i.e., the most accurate model target distributions) correspond to the models that have no target mode error ($m_b = m_{t*}$) and no base mode error ($m_{b\rightarrow} = m_b$) for $m_b = 4$.

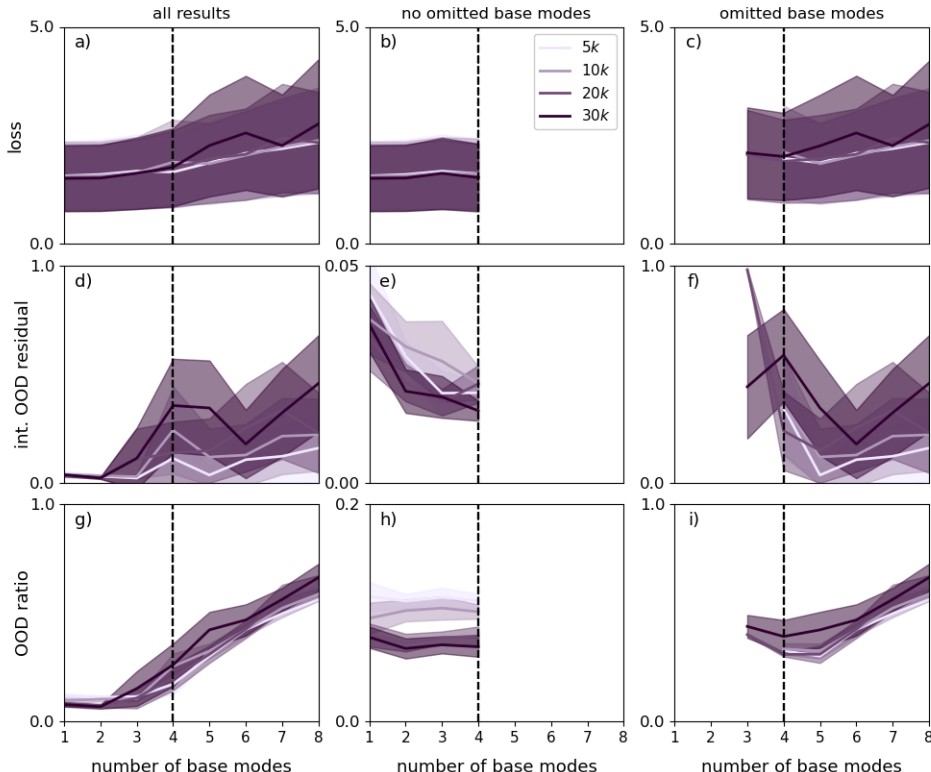

Figure 6: Three measures of model performance: loss (Equation 18), integrated OOD residual (Equation 21), and OOD ratio for four different training dataset sample sizes ($n_s = \{5, 10, 20, 30\} \cdot 10^3$). For each training dataset sample size 36 models were trained. The solid lines represent the mean of the 36 models and the shaded regions correspond to the standard deviation of the 36 models. $e$ shows that with no target mode error and no base mode error, the best model fit of the true target distribution has an integrated OOD residual value of about 0.025 for training dataset sizes. The vertical dashed line indicates the true target distribution number of modes, $m_{t*}$. The integrated OOD residual is the best representation of the quantity of OOD likelihood because it excludes the true target distribution and only quantifies OOD likelihood.

## 5 DISCUSSION

Prescribing multiple base distribution modes to match an unknown number of true target distribution modes is not a common practice when training normalizing flows. For high-dimensional problems, such as 256x256 pixel image datasets, this approach would be computationally expensive, as the number of true target distribution modes could be extremely large. It seems plausible that target mode error may be a source of significant OOD likelihood in high dimensional datasets. The effects of target mode error on high dimensional datasets is an open question for future research.

For small data problems, our empirical results suggest that both target mode error and base mode error can be eliminated by adjusting the number of base modes and using a simple criterion to ensure $m_{b\rightarrow} = m_b$ and $m_b = m_{t*}$ (i.e., the criterion of Equation 20), even when the number of true target distribution modes $m_{t*}$ is unknown (which is typically the case for real-world datasets). We have shown that both target mode error and base mode error can produce spurious OOD model likelihood. Therefore, the criterion can be used to improve model trustworthiness for small data imputation tasks by tractably eliminating two sources of spurious OOD likelihood.

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
