# OpenReview forum: "Flow-based imputation of small data"
_ICLR.cc/2025/Conference — Submitted to ICLR 2025_

### Official Review · Reviewer_A3rg · 2024-10-28

**Soundness:** 1
**Presentation:** 1
**Contribution:** 1
**Rating:** 3
**Confidence:** 3

**Summary:**

Normalizing flows assume a diffeomorphism exists from a base distribution to a target distribution. The authors note that this assumption implies that base and target distributions must have the same number of modes. Here, a mode is defined as a subset of the support of the distribution that is disconnected from the rest of the support.
The authors note that when using a standard normal distribution as a base distribution, this implies that spurious "filaments" between the true modes must exist to match the number of base modes (here, one) and transformed modes. They use a standard normalizing flow from Tensorflow on a 2D toy problem to investigate the effect of using a multimodal base distribution and how to select the number of base modes when the number of target nodes is unknown. They also introduce metrics, such as the Out-Of-Distribution (OOD) residual and ratios, to measure the performance of their method. The ultimate goal is to estimate the target distribution better to detect OOD samples.

**Strengths:**

The initial idea to notice that multi-modal distributions will inherently be ill-approximated by diffeomorphism is interesting and potentially solves a problem of the method. Trying to tackle this problem to perform data imputation, specifically for scientific issues, is an essential contribution.
The toy example is thoroughly explored with appealing figures.

**Weaknesses:**

The paper, and in particular section 2, lacks mathematical rigor. A mode is vaguely defined, while the Supp (support) is not defined. Moreover, since the paper decides to tackle approximate modes, the definition in equation 6 should have been modified to feature a tolerance epsilon to define approximate modes. And since approximate modes are considered, nothing stops a diffeomorphic map from mapping one mode to two approximately disjointed modes, provided that the filaments are sufficiently low probability in practice.  Thus, the conjecture (10) is more of a best-case scenario. Equation 21 is also not correctly mathematically defined and could be simplified using indicator functions.

The paper shows that some modes may be left unused, even when the total number of modes between base and target are equal. No solution is provided to correct this problem.

I find the experiment lacking on several fronts.
First, with a uniform target distribution and support made of squares, the diffeomorphism assumption is already violated, even accounting for the modes. This can be seen in the relatively poor recovery, which seems full of artifacts, even with the right number of base modes. It also seems to dominate the error, as the best OOD residual is with one base mode.
The introduction mentions applications in physical sciences, but the actual example is very toy-like. There is also no proof that it captures any of the characteristics of a real-world example.
The experiments seem to show that while having multiple modes for the base distribution would be better, this is a very risky endeavor. It risks leaving base modes unassigned, which results in high OOD errors.
Finally, the presentation lacks color bars to indicate the magnitude of the errors, making it difficult to evaluate the results visually.


The title, introduction, and discussion are somewhat misleading. The toy example does not reflect physical science applications very well. The high dimensionality needs to be tackled. Despite being in the title, the small data part is only mentioned in the conclusion and only as a consequence of the computational complexity. At the same time, none of the difficulties of working with a few samples have been considered.

**Questions:**

It also seems that the OOD ratio is simply a Monte-Carlo estimate of an integral similar to the OOD residual, so why not use an integral here as well?

Why would a real-world example feature multiple modes? How do we know it is an important source of error for OOD classification.

---

### Official Review · Reviewer_zmuV · 2024-10-29

**Soundness:** 2
**Presentation:** 3
**Contribution:** 2
**Rating:** 3
**Confidence:** 3

**Summary:**

The article investigates potential pathologies when using normalising flow methods for data imputation. Two chief sources of errors are explained and identified concerning the mismatch between the underlying number of assumed modes in the normalised space and the data generating distribution. These errors are demonstrated on a toy model, and methods are proposed to identify these issues and practice and avoid them. The integrated OOD residual is proposed as a metric for evaluating model success in an imputation context.

**Strengths:**

The explanation of the methods is clear and informative:

The examples listed in lines 31/32 are quite specific and niche, but not irrelevant.
The basic explanation of normalising flows is really clear and helpful.
The explanation of the two types of error from these models makes sense and sets up the rest of the article well.
Analysis of the toy target distribution is nice. The diagrams are very informative for the problems that you are trying to highlight and solve.
Integrated OOD residual is a reasonable metric to use for predictive accuracy.

**Weaknesses:**

Some notation could be changed, specifically the use of “m_{b->}”, which is a little too similar to m_b. Also m_t* vs m_t.
Do you really not have any applications beyond the four-squares toy example? This article is unconvincing without any real data or even vaguely realistic simulated data.
There is very little discussion or implementation specifically relevant to imputation as set up in the introduction. If the point of the article is about types of error when training normalising flows then the framing around imputation is unnecessary and confusing.

**Questions:**

Article framing:

Is low-dimensional imputation that difficult? What is the comparison with methodologically? Does this model just solve an easy problem with a greater (and not strictly necessary) degree of accuracy?
This article makes a couple of points well concerning potential pathologies of normalising flows and the underlying number of modes, but really doesn’t fulfil any initial promises concerning imputation questions. Does this method make a substantial difference when used in the context of a larger analysis pipeline of which imputation is a single component? What about when compared to other imputation methods, which are pretty effectively developed for low-dimensional problems already? The article gives no indication of this, while implying from the title onwards that this is going to be the point.
Does this article have much to say about predictive uncertainty of the normalising flow methods? Can something like multiple imputation be done effectively here, where the uncertainty of the imputation method can effectively be marginalised out? Do we expect these uncertainties to be well-calibrated against reality? Neural networks do not in general offer well-calibrated probabilistic predictions: this might be considered a major drawback in an imputation context.

Experiment design/rigour:

What is the computational setup? Would this work on a laptop? Do you need a server? TPUs?
Do you not have any applications beyond the four-squares toy example? Are you expecting people to apply this as an imputation method on real data without any more convincing experiments from your study?

Compute questions:

What’s the differing compute load between different sizes of training data or dimensionality of problem? Is the issue of scaling to higher dimensions simply a case of computational cost or more fundamental?
“The following architectural details and hyperparameters were optimised for 2D toy distributions by trial and error” - more details here would be good. How much work did this really comprise? How much extra computation? Could this be done systematically or was it done by hand? Similarly for batch sizes/epochs/learning rates/regularisation coefficients. What was your metric for success when tuning?

---

### Official Review · Reviewer_XcLf · 2024-10-30

**Soundness:** 2
**Presentation:** 1
**Contribution:** 2
**Rating:** 3
**Confidence:** 2

**Summary:**

This paper explores the use of normalizing flows for imputing incomplete small
datasets. It focuses on the diffeomorphism assumption and identifies two key
types of errors: target mode error and base mode error. Overall, the
presentation is convoluted, making the core contributions difficult to follow.

**Strengths:**

The paper explores an underexplored technique of data imputation using
normalizing flows, making it a novel application.

**Weaknesses:**

1. The paper lacks a theoretical analysis of its proposed approach.

2. The presentation needs significant improvements. The following points are
   indicative rather than exhaustive:
   - Key terms and notations should be defined and explained. For example,
     "out-of-distribution" is a key term for the paper and can be explained in
     one or two sentences, but this is not done in the paper.
   - The introduction to normalizing flows needs improvements. Equation (1) is
     simply a change of variables. Its connection to flows needs to be
     clarified. What do you mean by "approximately satisfy"? If $p_x()$ is a
     density function, is the value of (1) one? Is the main goal of normalizing
     flows to learn a transform $f$ such that $f^{-1}(x)$ follows a Gaussian
     distribution? What is the purpose of that? Also, what do you mean by
     "target sample"?
   - Do the "mode(s)" discussed before equation (6) have the same meaning as the
     "mode" defined in line 139? In any case, "mode" has a specific definition
     in probability distributions, so it is better to use another term.
   - Some equations, such as (8) and (9), do not contribute meaningfully to the
     presentation. What do you mean by "finitely bounded"?
   - Does ReLU satisfy the three conditions for diffeomorphisms?

**Questions:**

Please see Weaknesses.

---

### Meta-Review · Area_Chair_emnU · 2024-12-19

**Metareview:**

The reviewers have raised several concerns about lack of sound theoretical motivation, lack of proper exposition, and lack of experimental rigor during the review stage. the authors have chosen not to rebut, which is why i am recommending rejection for the paper.

**Additional Comments On Reviewer Discussion:**

THere was no rebuttal from the authors, and the reviewers were generally in agreement. so there was no discussion needed

---

### Decision · Program_Chairs · 2025-01-22

Reject